# Exposure to 16 h of normobaric hypoxia induces ionic edema in the healthy brain

Armin Biller [1,7 ✉], Stephanie Badde [2,3,7], Andreas Heckel[1], Philipp Guericke[4], Martin Bendszus[1], Armin M. Nagel [5,6], Sabine Heiland[1], Heimo Mairbäurl[4], Peter Bärtsch[4] & Kai Schommer[4]

Following prolonged exposure to hypoxic conditions, for example, due to ascent to high altitude, stroke, or traumatic brain injury, cerebral edema can develop. The exact nature and genesis of hypoxia-induced edema in healthy individuals remain unresolved. We examined the effects of prolonged, normobaric hypoxia, induced by 16 h of exposure to simulated high altitude, on healthy brains using proton, dynamic contrast enhanced, and sodium MRI. This dual approach allowed us to directly measure key factors in the development of hypoxia-induced brain edema: (1) Sodium signals as a surrogate of the distribution of electrolytes within the cerebral tissue and (2) $K^{trans}$ as a marker of blood–brain–barrier integrity. The measurements point toward an accumulation of sodium ions in extra- but not in intracellular space in combination with an intact endothelium. Both findings in combination are indicative of ionic extracellular edema, a subtype of cerebral edema that was only recently specified as an intermittent, yet distinct stage between cytotoxic and vasogenic edemas. In sum, here a combination of imaging techniques demonstrates the development of ionic edemas following prolonged normobaric hypoxia in agreement with cascadic models of edema formation.

[1] MDMI Lab, Department of Neuroradiology, University of Heidelberg, Heidelberg, Germany. [2] Department of Psychology, Tufts University, Medford, MA, USA. [3] Department of Psychology, New York University, New York, NY, USA. [4] Department of Sports Medicine, University of Heidelberg, Heidelberg, Germany. [5] Medical Physics in Radiology, German Cancer Research Centre (DKFZ), Heidelberg, Germany. [6] Institute of Radiology, University Hospital Erlangen, Friedrich-Alexander-Universität Erlangen-Nürnberg (FAU), Erlangen, Germany. [7] These authors contributed equally: Armin Biller, Stephanie Badde. ✉email: armin.biller@med.uni-heidelberg.de

Prolonged exposure to hypoxic conditions can lead to the development of cerebral edema. Otherwise healthy humans can encounter hypoxia under a variety of neurological conditions such as ischemic stroke or traumatic brain injury but also following ascend to high altitude. With increasing altitude, barometric pressure as well as partial pressure of atmospheric oxygen fall[1,2]. As a consequence, partial pressure and total content of arterial oxygen decline, less oxygen reaches the tissue, and hypoxia sets in. In response, cerebral blood flow increases to maintain oxygen delivery to the brain. Although this and other adaptive mechanisms are initiated, symptoms of acute mountain sickness (AMS) may evolve. Headache is cardinal, but symptoms also include loss of appetite or nausea, dizziness, fatigue or lassitude, and insomnia[3,4]. Additionally, high-altitude cerebral edema (HACE)—a life threatening condition—can develop. The exact nature of hypoxia-induced edemas as well as their relation to AMS remain unresolved. Here, a combination of magnetic resonance imaging (MRI) techniques is used to precisely characterize cerebral edemas after 16 h of normobaric hypoxia. Based on our analysis of the literature (Table 1) and predictions derived from cascadic models of edema formation[5,6] (Fig. 1) we hypothesized the presence of ionic edemas. Ionic edema is a relatively recently defined type of extracellular edema that had not been demonstrated in healthy humans and cannot be identified based on canonical proton MRI alone.

Edema evolution is supposed to progress in stages, driven by osmotic gradients and endothelial permeability[5,6] (Fig. 1): First, intracellular cytotoxic edemas occur, these are followed by extracellular ionic as well as vasogenic edemas, and finally by hemorrhagic transformation. Cytotoxic edema of brain tissue is induced by deleterious events such as hypoxia; it is defined as accumulation of extracellular ions and water in the cells. As the additional intracellular fluid is drawn from the surrounding extracellular space, the amount of fluid within the interstitial compartment is preserved. Consequently, cytotoxic edema is not associated with brain swelling. Cytotoxic edema advances the development of ionic edema by generating an osmotic driving force that pulls water from the intravascular compartment into interstitial brain tissue. Additional tissue water results in brain volume increase, which in combination with an intact blood–brain–barrier is a defining characteristic of ionic edema. If the deleterious influence persists, disruption of blood–brain–barrier integrity sets in; a permeability pore is formed, which in turn leads to the formation of vasogenic edema. Further damage of the endothelium results in growing permeability pores and the passage of erythrocytes into the brain tissue, i.e., hemorrhagic transformation.

This cascadic model of edema formation (Fig. 1) explains the heterogenic outcomes of previous canonical, proton MRI studies investigating the impact of normobaric hypoxia on the healthy brain (Table 1). Studies in which exposure to hypoxic conditions comparable to 4500 m was limited to 2–10 h found no evidence for brain volume changes[7–9] or cerebral edema[9] at the end of the exposure period. In contrast, several other studies with only slightly longer exposure periods[10–13] or higher simulated altitudes[14] report changes in brain volume and either intracellular edema[7,15], extracellular edema[9,14] or both[11,16]. Yet, a classification of edema based on canonical proton MRI is always based on indirect and incomplete evidence. Apparent diffusion coefficient (ADC) values, the most common marker, allow solely for a characterization of the mobility of water molecules and, thus, provide indirect information regarding the distribution of water molecules across intra- and extracellular space. The same holds for changes in T2 relaxation times. A direct classification of intra- and extracellular edema requires an assessment of $Na^+/K^+$-ATPase function. A characterization of the local mobility of electrolytes such as sodium ions hints at their distribution across the different cerebral fluid compartments and thus allows for an assessment of $Na^+/K^+$-ATPase function. In turn, a reliable distinction between ionic and vasogenic extracellular edema can only be achieved by a characterization of blood–brain–barrier integrity. Here we included both critical assessments combining sodium and dynamic contrast enhanced (DCE) MRI.

Sodium MRI enables non-invasive assessment of sodium ion mobility in brain tissue. Moreover, sodium MRI can yield both, average tissue sodium (ATS) and fluid-attenuated sodium (FAS) signals. The FAS signal is driven by sodium ions with short relaxation times. Consequently, a weighting towards the intracellular sodium compartment is achieved[17–20]. The conjoined analysis of ATS and FAS signals allows for a differentiation between brain areas with impaired $Na^+/K^+$-ATPase compatible with intracellular sodium accumulation, such as in focal ischemia[21–23], high-grade brain tumors[18,19,24–26], or acute multiple sclerosis lesions[27–30], and areas with extracellular sodium accumulation, such as in perifocal edema or chronic inflammatory lesions.

The cascadic development of cerebral edema depends on the permeability of the endothelium. Consequently, a characterization of blood-brain-barrier function is decisive for a precise classification of brain edema. DCE MRI allows for an assessment of gadolinium-enhanced signal kinetics. Based upon these signals, the transfer constant $K^{trans}$, a measure of the endothelial permeability, can be derived allowing for a non-invasive characterization of blood–brain–barrier function.

Here, we combined sodium and DCE proton MRI to achieve a precise characterization of the state of the edema formation after 16 h of hypoxic exposure. This exhaustive characterization of the consequences of normobaric hypoxia comprises changes in the distribution of electrolytes and water molecules within the cells, indicated by changes in the FAS signal, as well as in extracellular space, indicated by changes in the FAS signal, and the state of the blood–brain–barrier, indicated by changes in the transfer constant $K^{trans}$. This study therefore assesses the driving forces of edema formation. The parallel assessment of traditionally used proton MRI measures, the apparent diffusion coefficient (ADC) as well as brain tissue perfusion, allows us to relate the findings to previous and future studies on the effects of normobaric hypoxia on the healthy brain. Participants underwent two MR scans one before and one after they had spent 16 h under hypoxic conditions in a room with an atmospheric composition comparable to an altitude of 4500 m. In line with previous research, AMS severity scores were used to monitor individually experienced symptoms, and the corpus callosum, the white matter, and the nucleus lentiformis were defined as regions of interest (ROI; Fig. 2). To foreshadow the results, our measurements reveal the formation of ionic edemas, characterized by extracellular fluid accumulation in the presence of an intact blood–brain–barrier.

## Results

**Arterial oxygen content ($CaO_2$).** Our experimental protocol was successful in inducing hypoxemic hypoxia. During 16 h of exposure to low levels of atmospheric oxygen, partial pressure of oxygen in the blood, $pO_2$ significantly declined from on average 86.55 (0.91; standard error of the mean) to 34.98 (0.94) mmHg by 59% ($M_{diff} = 52$, $CI_{diff} = [49,54]$, $t(22) = 36.51$, $p < 0.001$, $d = 7.61$) and mean arterial oxygen, $CaO_2$ significantly dropped from on average 20.29 (0.26) to 15.28 (0.46) ml/dl by 25% ($M_{diff} = 5$, $CI_{diff} = [4,6]$, $t(22) = 11.17$, $p < 0.001$, $d = 2.33$).

**Acute mountain sickness (AMS) symptoms.** At the end of the hypoxic exposure period, the majority of participants showed symptoms of AMS. In total 16 of the 23 tested individuals developed AMS, according to two different scales (Lake Louise

**Table 1 Summary of previous studies on normobaric hypoxia and cerebral edema.**

| Study | N | SpO$_2$ (%) | Duration (h) | Normobaric hypoxia | Hypobaric hypoxia | Simulated altitude (m) | Δ Whole brain | Δ Total gray matter | Δ White matter | Δ Basal ganglia | Δ Genu corp. call. | Δ Splen. corp. call. | Δ Cerebellum | Δ Ventricle/CSF | Brain volume changes | Intra-cellular edema | Extra-cellular edema | Measure[4] |
|---|---|---|---|---|---|---|---|---|---|---|---|---|---|---|---|---|---|---|
| Muza et al., 1998 | 11 | 78 | 32 | X | | 4572 | | X | X | | | | | | ⇑ | – | – | – |
| Morocz et al., 2001 | 9 | 75 | 32 | | X | 4572 | | X | X | | | | | | ⇑ | – | – | T2r |
| Fischer et al., 2004 | 10 | 83§ | 10 | | X | 4500 | X | | | | | | | X | ⇑ | – | – | T2WI, DWI$_{B1000}$ |
| Kallenberg et al., 2007 | 22 | 76 | 16 | X | | 4500 | | | X | X | X | X | X | | ⇑ | X[1] | X | T2r, DWI$_{ADC}$ |
| Schoonman et al., 2008 | 9 | 78 | 6 | X | | 4500 | | | X | | | | | X | – | X[2] | X | DWI$_{B0,ADC}$ |
| Mairer et al., 2012 | 20 | 74& | 8 | X | | 5500 | | X[5] | X[5] | | | | | | ⇑[6] | – | X | DWI$_{ADC}$ |
| Lawley et al., 2013 | 13 | 81 | 2 | | | 4500 | | | X | X | X | X | X | | ⇓⇑ | X | – | T2r, DWI$_{MD,FA}$ |
| | | 81 | 10 | | | | | | X | X | X | X | X | | ⇑ | X | – | T2r, DWI$_{MD,FA}$ |
| Hunt et al., 2013 | 18 | 83 | 48 | | X | 3800 | | | X | X | X | X | | | – | X[2,3] | – | DWI$_{B0,ADC}$ |
| | | 87 | 168 | | | | | | X | X | X | X | | | – | X[2,3] | – | DWI$_{B0,ADC}$ |
| Lawley et al., 2014 | 13 | 81 | 2 | X | | 4572 | X | X | X | | X | X | | X | ⇓⇑ | – | – | DWI$_{ADC}$ |
| | | 81 | 10 | | | | X | X | X | | X | X | | X | ⇑ | – | – | DWI$_{ADC}$ |
| Sagoo et al., 2017 | 12 | 83 | 2 | X | | 4400 | | X | X | | X | X | | X | ⇓⇑ | – | – | DWI$_{ADC}$ |
| | | 83 | 4 | | | | | X | X | | X | X | | X | ⇓⇑ | – | – | DWI$_{ADC}$ |
| | | 84 | 6 | | | | | X | X | | X | X | | X | ⇓⇑ | – | X | DWI$_{ADC}$ |
| | | 84 | 11 | | | | | X | X | | X | X | | X | ⇓⇑ | – | X | DWI$_{ADC}$ |
| | | 85 | 22 | | | | | X | X | | X | X | | X | ⇑ | – | | DWI$_{ADC}$ |

Δ: Changes over time.
T2r: T2-relaxation times, T2WI: T2-weighted imaging using T2-TSE signals, DWI: diffusion-weighted imaging, ADC: apparent diffusion coefficient, MD: mean diffusivity (= isotropic ADC), FA: fractional anisotropy, $B_0$: non-diffusion weighted echo-planar image, $B_{1000}$: diffusion-weighted imaging with b = 1000 s/mm$^2$.
§ average across subgroups.
& data were assessed during passive and active hypoxia (active hypoxia included physical exercises); value refers to passive hypoxia.
[1] SCC only.
[2] AMS pos.
[3] except for SCC.
[4] these measures are surrogates of tissue fluid (T2) and mobility (DWI) and can point towards compartmental shifts but not reliably differentiate between cytotoxic, ionic and vasgogeneic edema.
[5] results are based on voxel-based morphometry rather than ROI-based volume estimations.
[6] MR measurements were performed under normoxia starting about 15 min after cessation of hypoxic conditions.

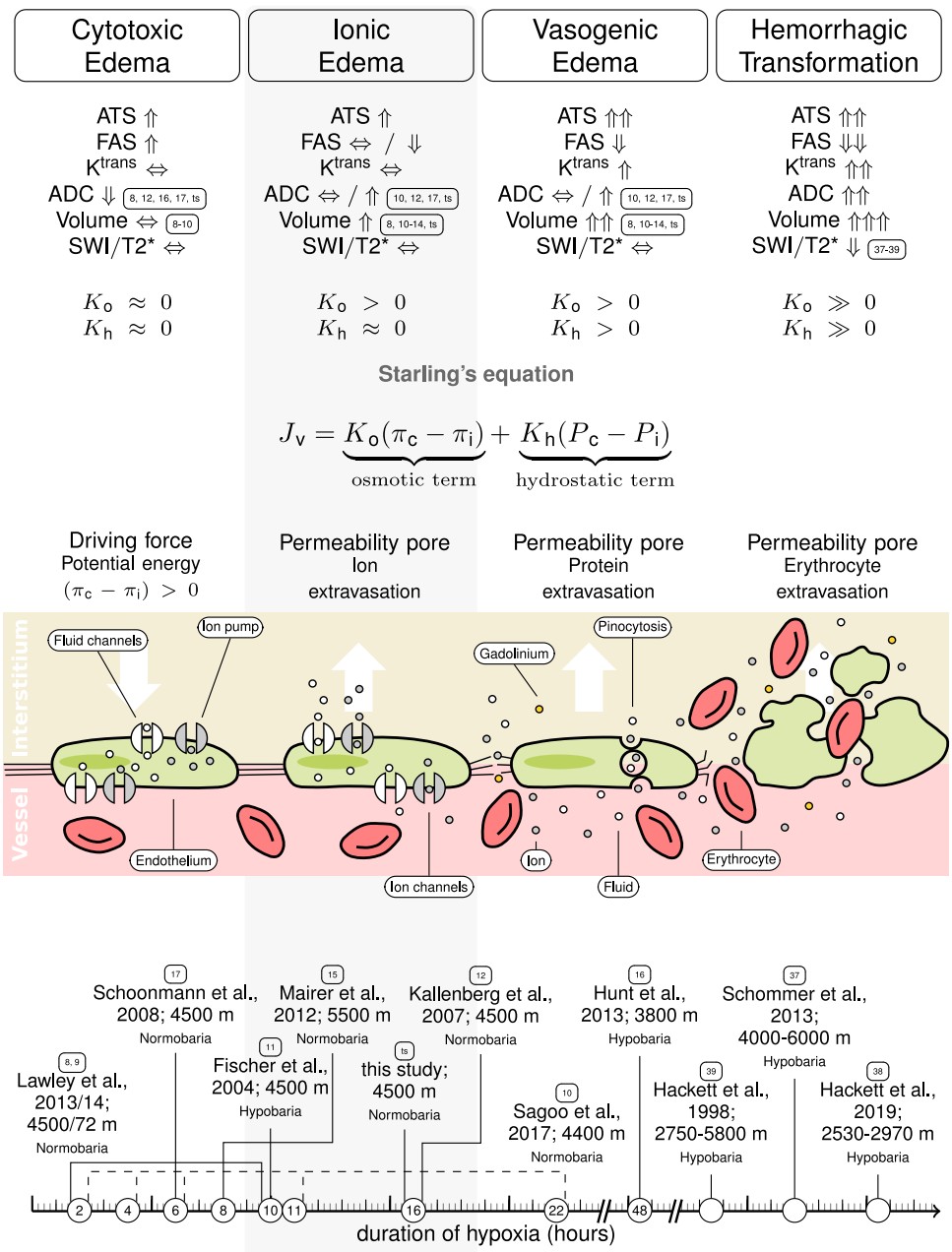

**Fig. 1 Stages of cerebral edema formation.**[6]. In cascadic models of edema formation the central driving forces are derived from Starling's equation[5,31]: Water flux ($J_v$) across the endothelium depends on capillary ($\pi_c$) and interstitial osmotic pressure ($\pi_p$), the osmotic conductivity coefficient ($K_o$), as well as capillary ($P_c$) and interstitial hydrostatic pressure ($P_i$) and the hydraulic conductivity coefficient ($K_h$). (**1**) Cytotoxic edema is characterized by intracellular water and sodium ion accumulation ($\pi_c$-$\pi_i$ > 0). Fluid shifts are limited to the interstitium, and the blood–brain–barrier is intact. Cytotoxic edema is indicated by increased average tissue sodium (ATS) and fluid-attenuated sodium (FAS), neutral transfer constant $K^{trans}$, reduced apparent diffusion coefficient (ADC) values, and constant brain volume and susceptibility-weighted imaging (SWI) or T2* values compared to baseline. (**2**) Ionic edema is characterized by ion extravasation and extracellular fluid accumulation; the blood–brain–barrier is intact. In terms of Starling's equation, $K_o$ is elevated while the hydrostatic term is around zero. Ionic edemas are consistent with increased ATS, neutral or reduced FAS, constant $K^{trans}$, neutral or increased ADC, SWI or T2*, and increased brain volume. (**3**) Vasogenic edema is determined by ion and protein extravasation, extracellular fluid accumulation due to an impaired blood–brain–barrier. According to Starling's equation vasogenic edema is defined mainly by the hydrostatic term. Vasogenic edema is consistent with increased ATS, reduced FAS, increased $K^{trans}$, neutral or increased ADC, SWI or T2*, and increased brain volume. The latter three indicators are nearly identical to (2) rendering it practically impossible to differentiate between vasogenic and ionic edema based solely on canonical proton MRI. (**4**) Hemorrhagic transformation is characterized by further blood–brain–barrier disruption, which allows erythrocytes to pass into the interstitium. In Starling's equation, the hydrostatic term increases further. Hemorrhagic transformation is indicated by increased ATS, reduced FAS, increased $K^{trans}$, increased ADC, increased brain volume, and reduced SWI or T2* signal. The transition between edema types is fluid and might occur at different rates in different brain areas. The timeline at the bottom of the figure sorts previous studies according to the duration of hypoxic exposure. The measurements reported in these studies agree with our predictions as indicated by the numbered labels.

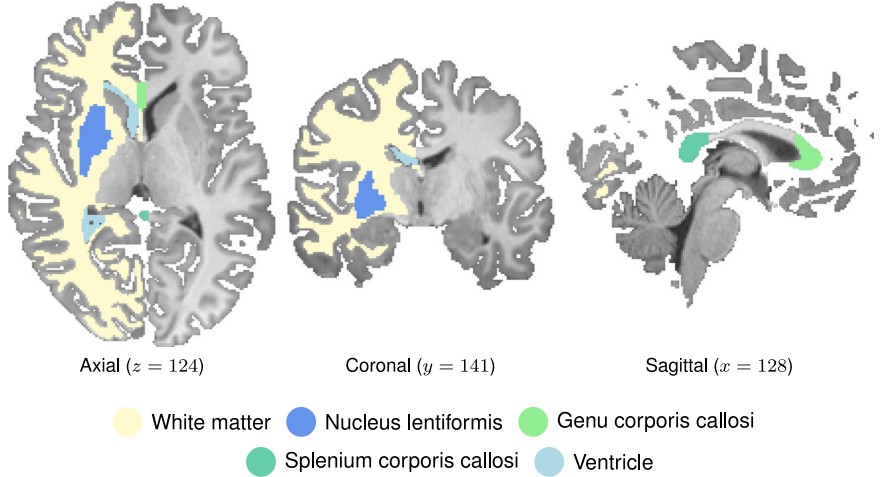

Axial ($z = 124$)  Coronal ($y = 141$)  Sagittal ($x = 128$)

- ◯ White matter
- ◯ Nucleus lentiformis
- ◯ Genu corporis callosi
- ◯ Splenium corporis callosi
- ◯ Ventricle

**Fig. 2 Regions of interest (ROIs).** Axial, coronal, and sagittal slices showing the color-coded ROIs for white matter, nucleus lentiformis, genu and splenium corporis callosi, and ventricles overlaid onto the right hemisphere (radiologic convention) of anatomical T1-weighted images.

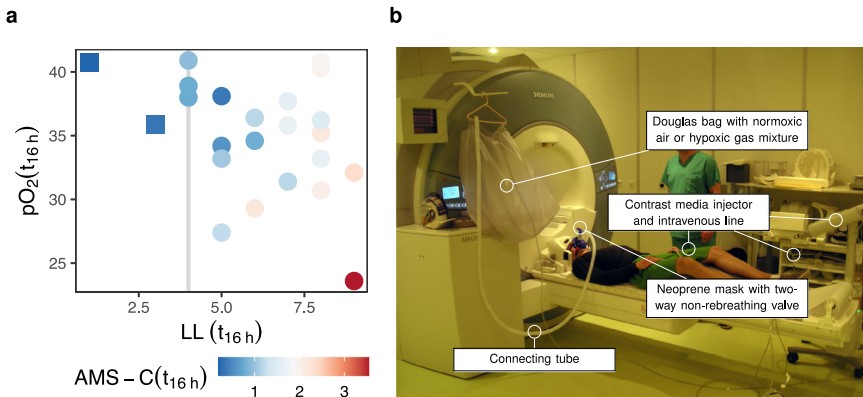

**Fig. 3 Exposure to low atmospheric oxygen and acute mountain sickness (AMS). a** Single participants' arterial oxygen levels ($pO_2$) as a function of their Lake Louise (LL) scores, a clinical measure of AMS, both measures are taken after 16 h of hypoxic exposure. Scores equal or above 4 indicate the presence of AMS. Marker colors correspond to AMS-C scores, a second clinical measure of AMS, scores above 0.7 indicate the presence of AMS. **b** To induce normobaric hypoxia, participants were exposed to an atmosphere corresponding to that at 4500 m altitude for 16 h. During MR measurements, participants wore a neoprene mask with a two-way non-rebreathing valve connected to a Douglas bag. Under normoxia conditions, the bag was filled with ambient air; under hypoxia conditions the bag was filled with a hypoxic gas mixture.

(LL) score > 4, AMS-C score > 0.7; Fig. 3a); only two participants fulfilled neither of the two criteria. LL- and AMS-C scores after 16 h of exposure to reduced atmospheric oxygen were highly correlated to each other ($r = 0.84$, CI = [0.65,0.93], $p < 0.001$), and either score correlated (marginally) significantly with $pO_2$ after 16 h of hypoxic exposure (LLS: $pO_2$ ($t_{16}$), $r = -0.39$, CI = [−0.69,0.02], $p = 0.066$, AMS-C: $pO_2$ ($t_{16}$), $r = -0.54$, $CI = [−0.78, −0.17]$, $p = 0.007$).

7 of 23 participants received antiemetic medication (Motilium 2 ml). Due to headache, 12 of 23 participants required analgesic medication (Ibuprofen 400 mg) at least once (1–3 times), 2 of these participants took the drug within 2 h before the second MRI scan; the remaining 10 participants received analgetic medication at least 6 but typically around 10–12 h before the second MRI scan (Supplementary Table 1). Consistently, at the time of the second MR scan, at the group level, analgesic medication was not anymore associated with significantly reduced AMS symptoms (LL-scores: $M = 6.58$, $CI = [5.21,7.95]$ with analgetic treatment and $M = 5.64$, $CI = [4.43,6.84]$ without treatment; Supplementary Table 2). There was no significant effect of Ibuprofen on the measured changes in brain volume, perfusion, cerebral oxygen delivery, diffusion, sodium signals, or the transfer constant in any ROI (Supplementary Table 2).

**Brain volume.** Morphometric analyses revealed significant increases in volume after hypoxic exposure for white matter, nucleus lentiformis as well as genu corporis callosi (Fig. 4, Supplementary Table 3). In contrast, ventricle volume decreased significantly, and this decrease was negatively correlated with the volume increases of the nucleus lentiformis (Fig. 4, Supplementary Fig. 3, Supplementary Table 4).

Volume changes in the nucleus lentiformis correlated significantly with ATS signal changes and $pO_2$ ($t_{16}$). In turn, $pO_2$ ($t_{16}$) correlated negatively with volume changes of the genu corporis callosi and the ventricles (Fig. 4, Supplementary Figs. 2 and 3, Supplementary Table 5). Brain volume changes after exposure to hypoxic conditions across all ROIs did not correlate significantly with AMS scores (Supplementary Table 5).

**Average tissue sodium (ATS) signal.** ATS signal strength increased during hypoxic exposure in all ROIs (Fig. 4, Supplementary Table 3). Except for the corpus callosum (genu and splenium), regional ATS signal changes were mutually correlated (Fig. 4, Supplementary Fig. 2, Supplementary Table 4).

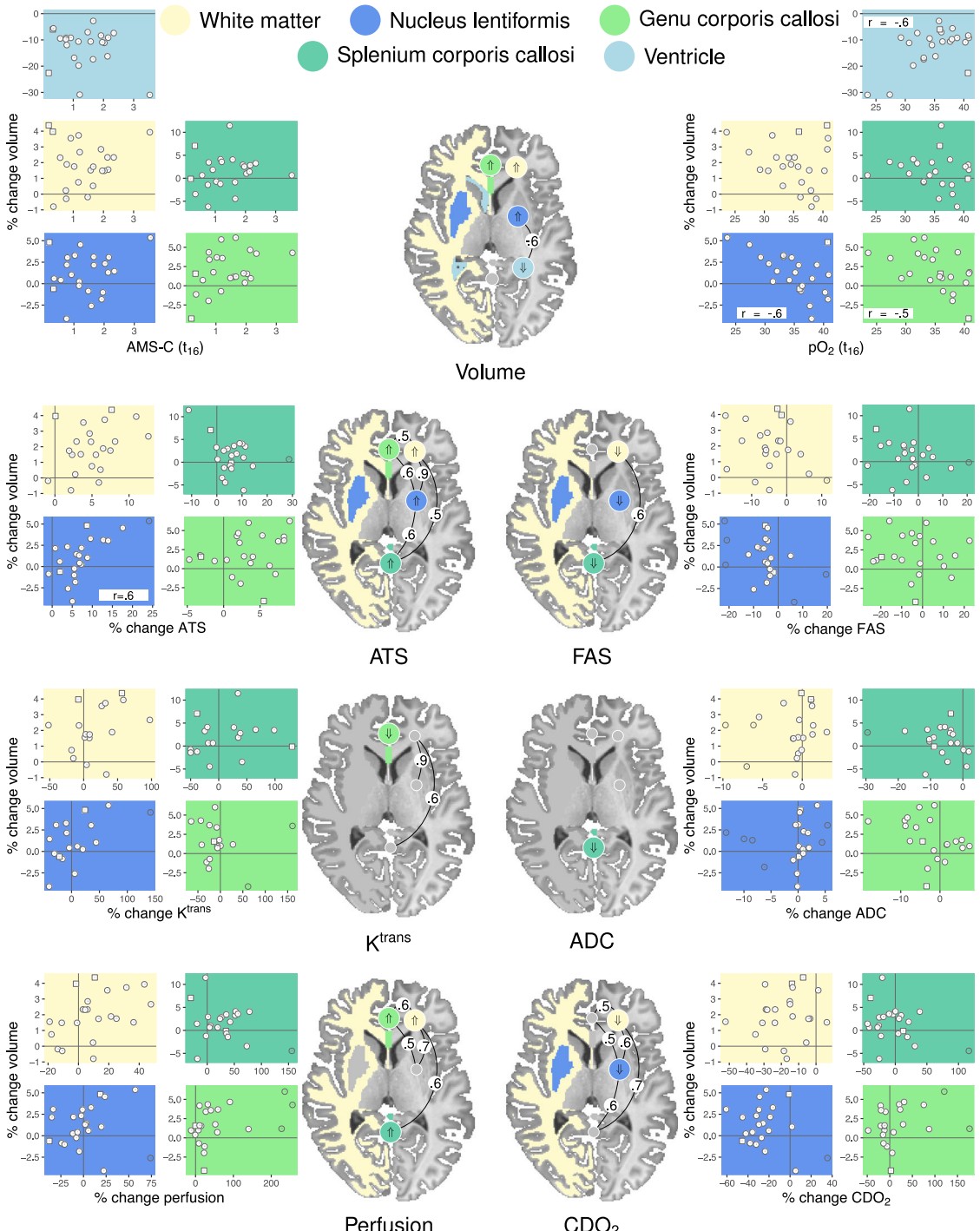

**Fig. 4 Hypoxia-induced regional, normalized changes in the brain after 16 h of hypoxic exposure.** Changes of brain volume, average tissue sodium (ATS) and fluid-attenuated sodium (FAS) signals, the transfer constant K$^{trans}$, apparent diffusion coefficient (ADC), perfusion, and cerebral oxygen delivery (CDO$_2$) values as well as altitude mountain sickness (AMS) scores are shown. For each measure, colored region of interest (ROI) overlays in the right hemispheres (radiologic convention) indicate changes after hypoxic exposure, the correspondingly colored circles in the left hemisphere indicate the direction of the change. Gray ROI overlays and small gray circles represent ROIs without significant change after hypoxic exposure in the respective signal. Significant correlations between ROIs are projected onto the left hemisphere of the brain (black arrows, white markers). Insets at the sides show normalized changes in each measure (x-axis) plotted against normalized changes in volume for reference. ROIs are color-coded. Correlation coefficients are reported only if these were significant. Filled circles represent data points included into the statistical analysis; open circles represent outliers excluded from the analysis.

Changes of ATS signal and cerebral blood flow correlated in the white matter and the nucleus lentiformis. ATS signal and K$^{trans}$ changes positively correlated in white matter; either signal change correlated negatively in the genu corporis callosi (Supplementary Fig. 3, Supplementary Table 5). There were no

significant correlations between ATS signals and pO$_2$ (t$_{16}$) or AMS measures (Supplementary Table 5).

**Fluid attenuated sodium (FAS) signal.** FAS signal strength decreased in white matter, nucleus lentiformis, and splenium

corporis callosi (Fig. 4, Supplementary Table 3). Changes in regional FAS signal strength in white matter and splenium corporis callosi were mutually correlated (Fig. 4, Supplementary Fig. 2, Supplementary Table 4). There were no significant correlations between FAS signals and $pO_2$ ($t_{16}$) as well as changes in AMS measures or other neurophysiological variables (Supplementary Fig. 3, Supplementary Table 5).

**Transfer constant ($K^{trans}$).** No significant increase in $K^{trans}$ values was observed after 16 h of hypoxic exposure (Fig. 4, Supplementary Table 3) independently of the pharmacokinetic model used to estimate $K^{trans}$. Regional $K^{trans}$ values were correlated between the white matter and nucleus lentiformis as well as between the white matter and the splenium corporis callosi (Fig. 4, Supplementary Fig. 1, Supplementary Table 4). Also, $K^{trans}$ values and perfusion changes positively correlated in the white matter (Supplementary Fig. 2, Supplementary Table 5). There were no significant correlations between $K^{trans}$ and $pO_2$ ($t_{16}$) or AMS measures (Supplementary Table 5).

**Apparent diffusion coefficient (ADC).** After 16 h under hypoxic conditions, ADC values were reduced in the splenium corporis callosi (Fig. 4, Supplementary Table 3). Regional changes did not significantly correlate across ROIs (Fig. 4. Supplementary Fig. 1, Supplementary Table 4). There were no significant correlations between changes in the ADC and $pO_2$ ($t_{16}$), changes in other neurophysiological variables or AMS measures (Supplementary Fig. 2, Supplementary Table 5).

**Brain tissue perfusion.** Hypoxic exposure was associated with an increase in cerebral blood flow in the white matter as well as in the genu and splenium of the corpus callosum (Fig. 4, Supplementary Table 3). Perfusion changes correlated between genu corporis callosi and white matter, between white matter and nucleus lentiformis, between genu corporis callosi and nucleus lentiformis, and between white matter and splenium corporis callosi (Fig. 4, Supplementary Fig. 1, Supplementary Table 4). Changes in brain tissue perfusion correlated with ATS signal changes in the white matter and the nucleus lentiformis and with $CDO_2$ changes in all ROIs (Supplementary Fig. 2, Supplementary Table 5). There were no significant correlations between changes in perfusion and $pO_2$ ($t_{16}$) or AMS measures (Supplementary Table 5).

**Cerebral oxygen delivery ($CDO_2$).** During hypoxic exposure, regional $CDO_2$ significantly declined in the white matter and nucleus lentiformis (Fig. 4, Supplementary Table 3). Except for the corpus callosum (genu and splenium), regional $CDO_2$ changes were mutually correlated (Fig. 4, Supplementary Fig. 1, Supplementary Table 4). In addition, $CDO_2$ changes were correlated with perfusion changes in all ROIs (Supplementary Fig. 2, Supplementary Table 5). There were no significant correlations between $CDO_2$ changes and AMS measures (Supplementary Table 5).

## Discussion

Prolonged exposure to hypoxic conditions can lead to the formation of cerebral edema. To comprehensively assess the nature of these edemas, we exposed healthy humans to 16 h of normobaric hypoxia created by simulating a sudden ascent to a high altitude of 4500 m and measured the effects on the brain using sodium, DCE, and proton MRI techniques. Using these measures, we characterized the distribution of brain tissue electrolytes as well as the state of the blood brain barrier after prolonged normobaric hypoxia. Blood oxygen levels confirmed the sustained

presence of hypoxia, the majority of participants developed AMS symptoms, and proton MRI revealed accompanying but uncorrelated increases in cerebral volume. Our combined analysis of electrolytes and endothelial permeability attributed the observed tissue changes after prolonged hypoxia to the presence of extracellular, ionic edema. Ionic edema is a subtype of cerebral edema that was only recently specified as an intermittent, yet distinct stage between cytotoxic and vasogenic edemas, and cannot be differentially diagnosed based on canonical proton MRI measures. Thus, our study on the effects of normobaric hypoxia on the brain provides in vivo evidence for cascadic models of edema formation[5,6,31,32] using non-invasive MRI.

**Responses of the human brain to hypoxia.** The brain responded to hypoxic conditions with increased cerebral blood flow in white matter, genu and splenium corporis callosi—a known physiological response to hypoxic conditions[33,34] which reflects the body's attempt to protect the brain from hypoxia-induced damage. However, increased perfusion could not prevent a reduction in cerebral oxygen delivery in the white matter and the nucleus lentiformis. As a result, hypoxia led to volume increases in all regions of interest and a compensatory volume reduction of the ventricles as well as the formation of brain edema.

Our combined measurements suggest the presence of extracellular ionic edema after 16 h of hypoxic exposure: The combination of ATS signal increases and FAS signal decreases indicates extracellular interstitial sodium ion accumulation. Brain volume changes provide evidence for fluid shifts from the intravascular to the interstitial compartment. Yet, the brain–blood–barrier is still intact as indicated by stable $K^{trans}$ values.

**Cascadic development of hypoxia-induced cerebral edema.** We combined sodium and DCE MRI to directly asses two factors pivotal to edema formation according to Starling's equation[5,31]: (1) The osmotic driving force, which we can reconstruct from the distribution of electrolytes across extra- and intracellular compartments and (2) the permeability pore, which we assessed by inferring $K^{trans}$, an indicator of blood–brain–barrier permeability. These two forces lead to shifts of fluid from the vessels into the cerebral tissue, i.e., edema formation. Our approach in combination with a classification of previously published results acquired with canonical proton MRI enables us to suggest a comprehensive, empirically supported description of edema formation under hypoxic conditions.

*Intracellular, cytotoxic edema.* After limited exposure to hypoxic conditions, cytotoxic edema can develop (Fig. 1, 1st column). Cytotoxic edemas are characterized by an accumulation of electrolytes within the cells. This ion accumulation creates an osmotic gradient across the cell membrane and consequently leads to increased intracellular fluid accumulation. In cytotoxic edema, fluid shifts are limited to the parenchyma, there is no flux across the intact endothelium, hence, brain volume should remain stable. Previous studies are consistent with the presence of cytotoxic edema in individuals exposed to 2–10 h of hypoxic conditions; in these studies, canonical proton MRI demonstrated decreased ADC values in almost all regions of interest (Fig. 1, Table 1)[7,16]. Such a decrease in ADC is indicative of restricted mobility of water molecules which likely results from their intracellular accumulation[7,16] and thus likely indicates cytotoxic edema. Further in agreement with cytotoxic edema, no significant changes in brain volume emerged after only two hours of hypoxic exposure[7]. Even though the validity of a reduction in ADC as a marker of cytotoxic edema is well established[35], corroborative evidence in form of increased FAS and ATS signals after short

periods of hypoxic exposure could strengthen the conclusion in future studies.

The here reported measurements indicate—in agreement with cascadic models of edema formation[5,6,32]—that after prolonged exposure to hypoxic conditions cytotoxic edemas resolve as part of the development of ionic edema. Here, FAS signals provided no evidence for an accumulation of sodium ions within the cells indicating the absence of cytotoxic edema after 16 h of hypoxic exposure. Further evidence for the absence of cytotoxic edema in our study is provided by the presence of volume changes in nearly all areas of interest. Similarly, Kallenberg and colleagues found reduced ADC values after 16 h of hypoxia in the genu corporis callosi only[11]. Previous studies report increases in brain volume following approximately 8–10 h of hypoxic exposure[7,8,10], suggesting that the transition from cytotoxic to ionic edema begins after a few hours of hypoxia.

*Ionic edema.* The accumulation of fluid and ions in intracellular space during cytotoxic edema depletes these elements from extracellular space which in turn creates an osmotic and chemical gradient across the blood brain barrier. In the subsequent genesis of ionic edema (Fig. 1, 2nd column), fluid and ions move across the still intact endothelium into extracellular space. The consequences are cerebral swelling as well as increases in extracellular ion concentrations. In turn, due to the accumulation of water molecules in the interstitial compartment, their mobility indicated by ADC values and T2 relaxation times should be similar to baseline conditions. Crucially, ionic edema constitutes only the first stage of endothelial dysregulation and thus is associated with a still intact blood brain barrier.

Our measurements indicate the presence of ionic, extracellular edema after 16 h of hypoxia. The combination of sodium signals (ATS and FAS, Fig. 1) indicated extracellular sodium accumulation in our study. Moreover, the observed volume increase (brain volume, Fig. 4) provides evidence for fluid shifts from the vascular to the interstitial compartment. Thus, the findings of the present study provide evidence for the evolution of extracellular ionic edema after 16 h of hypoxic exposure. Indirect corroborative evidence comes from previous studies with prolonged durations of exposure. These studies found increases in brain volume combined with increased T2 signals[9,11], respectively. Noteworthy such findings have traditionally been interpreted as evidence for vasogenic edema. Yet, newer models of edema formation[5,6] link vasogenic edema to disruptions in blood–brain–barrier integrity, an important criterion that has not been evaluated in previous studies. Thus, there is a possibility that previous studies with prolonged durations of hypoxic exposure, actually observed ionic rather than vasogenic edema. This assumption is in accordance with previous reports about the concurrent presence of intracellular edema – indicated by restricted diffusion[7,11]—and extracellular edema—indicated by increases in cerebral volume. Cascadic models of edema formation based on Starling's equation imply transition phases between the different edema types, and thus predict a temporary co-existence of cytotoxic and ionic edema.

*Vasogenic edema.* Our study revealed an intact blood–brain–barrier in all regions of interest, ruling out the presence of vasogenic edema. As discussed above, this finding raises the possibility that previous studies reporting vasogenic edema based on brain volume increases as well as T2-signal or ADC value elevations in the absence of measures of endothelial integrity point toward ionic edema formation after prolonged hypoxia, too. Thus, our results point towards an alternation of currently assumed time courses, the development of vasogenic edema under conditions of prolonged normobaric hypoxia might take considerably longer than previously assumed. During

vasogenic edema (Fig. 1, 3rd column), endothelial damage leads to disruption of the blood–brain–barrier, and fluid as well as plasma proteins pass from the intravascular to the interstitial extracellular compartment. In individuals surviving HACE, microhemorrhages were found in the splenium corporis callosi[36,37]. Thus, HACE seems to be characterized by blood–brain–barrier disruption suggesting that the typically massive cerebral volume changes are due to the occurrence of vasogenic edema[38] followed by hemorrhagic transformation.

**Regional specificity of hypoxia effects**. In the genu corporis callosi an unexpected result emerged, K[trans], which is anti-proportionally related to the strength blood–brain–barrier decreased under hypoxia when estimated using the Tofts model. Potentially, the observed change reflects a protective mechanism to sustain endothelial integrity in the presence of extracellular sodium ion accumulation. In agreement with the idea that an accumulation of sodium ions triggered this strengthening of the blood–brain–barrier, a negative correlation between K[trans] and ATS signal changes emerged in the genu corporis callosi. Consistent with a strong blood–brain–barrier, hypoxia induced no significant ADC or FAS signal changes in this area, i.e., no signs of intracellular water or sodium ion accumulation and depletion, respectively, in the genu corporis callosi.

**Relation between AMS symptoms and edema formation**. Headaches are the cardinal symptom of AMS which often leads to the characterization of hypoxia-induced edema as an escalation of untreated AMS. Yet, this link is under debate. Whereas some studies find a correlation between effects of hypoxia on the brain and AMS symptoms[7–9,11,16] others do not[10,13,15]. Our study points against a direct link between AMS and hypoxia-induced edema. The severity of AMS symptoms as assessed by LL- and AMS-C scores was independent of changes in brain volume and all other of the brain's responses to hypoxia. This missing statistical relationship might of course be due to a lack of statistical power, yet the pattern is consistent across measures of hypoxia-effects on the brain. Moreover, both scales of AMS symptoms, LL- and AMS-C scores, correlated strongly with blood oxygen levels. At the same time, volume changes in several brain areas correlated with partial oxygen pressure after prolonged hypoxic exposure, supporting direct links between the degree of hypoxia and the degree of AMS symptoms as well as between the degree of hypoxia and the development of cerebral edema. In sum, our data are in accordance with the possibility that prolonged hypoxia causes two, relatively independent processes, AMS symptoms and cerebral edema formation.

The current study tested the effects of prolonged normobaric hypoxia on the body and the brain. One might wonder how the results of the current as well as the majority of studies in high-altitude research (Table 1) translate to conditions of hypobaric hypoxia. There are small differences in the ventilatory and the renal response between exposures to normobaric and hypobaric hypoxia[39,40]. These differences may explain differences in fluid balance between normobaric and hypobaric hypoxia[40] since the isocapnic hypoxic ventilatory response, a maker of the sensitivity of the peripheral chemoreceptors, may be linked to the diuretic response in hypoxia[41]. More severe AMS following hypobaric hypoxia was found in small studies lasting 8–9 h[42,43] but not in larger studies involving 16 h of exposure[44]. Given the effects of hypobaria on the brain[45] we cannot finally exclude differences in the pathophysiology of cerebral edema as described here and under conditions of high altitude. This question has to be addressed experimentally. Our detailed characterization of the effects of normobaric hypoxia on the brain could provide a

valuable starting point to explore mediating influences of hypobaria. Importantly, the possibility that hypobaric conditions might accelerate the process of edema formation accentuates the importance of our observation that some participants showed massive effects of hypoxia on the brain, while experiencing hardly any symptoms of AMS.

**Medication effects**. About 50% of participants received analgetic medication (Ibuprofen). Most participants received their medication in the first hours of the study and only two participants took the analgetic drug within 6 h before the second MR scan. Consistently, at that timepoint there was no evidence for analgetic effects at the group level rendering it unlikely that the analgetic medication obscured a correlation between cerebral edema and AMS symptoms especially headaches. In theory, Ibuprofen could protect the blood–brain–barrier[46] by inhibiting cyclooxygenase-2, and possibly mitigating the upregulation of P-glycoprotein as has been shown for example in seizure-associated neuroinflammation[47]. However, statistical analyses revealed no significant effect of medication on cerebral volume changes, electrolyte distributions, and blood–brain–barrier integrity.

**Limitations of the current study and future research**. Even though our study covers the largest number of participants among MRI studies investigating the effects of hypoxia, the sample size is by far insufficient to draw conclusions about risk factors for hypoxia-induced cerebral edema. For this reason, we opted against an evaluation of predictors such as age, weight, and height. Instead, we recruited a homogenous sample, which in turn has the advantage of reduced inter-subject variance and increased statistical power. However, the size of our sample still limits the number of statistical tests and regions of interest that can be evaluated while keeping false discovery rates at a minimum. The sample size is sufficient to detect hypoxia-induced changes of an average effect size with a power of 0.8, but only large correlations can be detected with the same power. Thus, the interpretation of inter-area and inter-measure correlations underlies particular limitations compared to that of hypoxia-induced changes in the different measures evaluated here.

This study revealed the presence of ionic edema in healthy humans using a combination of novel and established MRI measures. Our analysis of the literature demonstrated near perfect agreement between our predictions based on cascadic models of edema formation and previously acquired proton MRI data after prolonged normobaric hypoxia. However, the measures used in previous studies provide only affirmative but not conclusive evidence and this study focuses on ionic edema, which previous studies failed to identify. Longer and longitudinal studies are required to characterize the final steps of escalation of the cascadic process up to hemorrhagic conversion as precisely as done here for the critical transition from intra- to extracellular edema. Due to increased risk, such studies pose additional challenges. The current study provides a blueprint for future studies (1) by establishing a comprehensive methodology to induce and monitor the development of hypoxia-induced cerebral edema and (2) by providing clear predictions for novel and established MRI measures based on Starling's equation.

In summary, our detailed measurements revealed increased cerebral perfusion, region-specific increases in brain volume, and extracellular accumulation of sodium ions in the presence of blood–brain–barrier integrity after 16 h of exposure to normobaric, hypoxic conditions. Using a combination of MRI techniques, we demonstrate the presence of extracellular, ionic edema in healthy humans following prolonged exposure to

normobaric hypoxia. Our results provide in vivo evidence for a cascadic evolution of brain edema according to Starling's equation, a theory which until now had been developed on theoretical grounds[5,31], on clinical observations and in animal models[5,6] In turn, a theoretical understanding of the mechanisms of edema formation provides a base for the development of treatments for the effects of hypoxia, for example, following stroke or traumatic brain injury, on the brain[6,48]. The current study demonstrates that the experimental protocols developed in high-altitude research in combination with multiple imaging techniques provide an excellent tool to study edema formation under hypoxia in theoretical and clinical contexts.

Moreover, our study ties into the literature on the development of cerebral edema under conditions of simulated high altitude, given prolonged hypoxia. The demonstration of ionic edema after prolonged normobaric hypoxia bridges the gap between reports of cytotoxic and vasogenic edema in previous studies while revealing the need for a precise characterization of edema type through an assessment of electrolyte distributions and blood–brain–barrier permeability in future studies. At the same time, our results highlight the complexity of the relation between AMS symptoms and hypoxia-induced cerebral edema and the need to differentiate between both in research and practice.

## Methods

### Protocol

*Study cohort*. Participants were recruited from the general population of Heidelberg, Germany (located 141 m above sea level) and medically examined prior to participation. Inclusion required absence of pathological findings after physical examination of the heart, lung, and arteries, standard laboratory blood parameters (creatinine, urea nitrogen, liver enzymes, ionic content, complete blood count), and a resting 12-lead-ECG. The final sample comprised 23 healthy, non-smoking, non-anaemic individuals (1 female; age: $25 \pm 3$ years; body mass index $22.6 \pm 1.6$ kg/m$^2$; height: $182 \pm 6$ cm). Four additional participants (2 females) did not consent to the second MRI scan after 16 h of exposure to hypoxic conditions citing different reasons (e.g., nausea, claustrophobia, insomnia during the exposure phase, …). Two further participants (both male, one middle author) aborted the study without even completing the first MRI scan. These participants' AMS symptom severity was not significantly different from that of the final sample (LL-scores sample mean 6, range 1–9; drop-out group: mean 5, range 4–7). For three additional participants (2 male) sodium MR-signals were off-resonant probably due to non-converging B0-shimming or insufficient flip angle adjustment. The final sample size of 23 participants allows for the detection of significant differences with an average effect size ($\bar{d}_z = 0.53$ at $\beta = 0.8$; determined with GPower[49]). None of the participants had been exposed to altitudes higher than 2000 m within a period of 30 days prior to the study. None of the participants took medications on a regular basis and none had a history of chronic headache. The study was conducted in accordance with the Declaration of Helsinki and its current amendments and was approved by the Ethics Committee of the Medical Department of the University of Heidelberg (protocol number S-463/2009). All participants provided written informed consent prior to participation.

**Consent to publish**. The authors affirm that human research participants provided written informed consent, for publication of the images in Fig. 3b.

*Experimental design and procedure*. Participants were exposed to normobaric hypoxia for 16 h. During this time, they remained in a room with an atmosphere composed of 12% $O_2$ and $N_2$ equivalent to the atmospheric composition at an altitude of 4500 m (System Linde Gas, Unterschleissheim, Germany) and an ambient temperature of 20–22 °C. Inspired $CO_2$ was kept below 0.3% by ensuring appropriate air flow. The exposure period included an over-night stay; participants received a standardized diet and beverages throughout the study.

To establish the effects of hypoxia on blood gas saturation, a capillary blood sample was taken from the ear lobe before ($pO_2(t_0)$) and at the end ($pO_2(t_{16})$) of the hypoxic exposure period (Siemens Rapidpoint 400/405, Bayer Diagnostics, Sudbury, UK). The severity of AMS was measured by means of the Lake Louise (LL) score[50] assessed immediately before and at the end of the hypoxic exposure period. LL scores combine a questionnaire with clinical assessment (scale: 0 = no symptoms, 1 = symptoms without influence on activity, 2 = reduced activity due to symptoms, 3 = forced to bed rest, and 4 = life-threatening symptoms). In addition, the AMS-C score, a sub-scale of the Environmental Symptom Questionnaire[51], was determined. AMS was affirmed if the following criteria were fulfilled; LL score > 4 and AMS-C score > 0.7. If required by the participant, headache was treated with NSAID (e.g., Ibuprofen, Paracetamol) and nausea or vomiting with antiemetics

(Domperidone) (see above and Supplementary Tabs. 1 and 2 for additional information on medication).

To measure the effects of hypoxic exposure on the brain, participants underwent two MR scans. A baseline MRI was performed within a period of 14 days before the study. The second scan was performed immediately after 16 h hypoxic exposure. During MR scans, participants wore a neoprene mask with a two-way non-rebreathing valve (Hans Rudolph, 2400 series, Kansas City, MO, USA). During the baseline scan, the inspiratory port of the mask was connected to a Douglas bag (250 l) that contained ambient, normoxic air (Fig. 3b). In preparation of and during the second MR scan, the inspiratory port of the mask was connected to a Douglas bag, which contained a hypoxic gas mixture of medical grade quality (12% $O_2$, balanced with $N_2$) that was delivered from pressurized cylinders.

## Magnetic resonance imaging

*Morphometry.* For morphometric analyses, an isotropic whole-brain T1-weighted MPRAGE sequence was acquired (details in Supplementary Table 6). Volumetric segmentation and longitudinal processing were performed using FreeSurfer (https://surfer.nmr.mgh.harvard.edu/; for details cf. Supplementary Methods).

Our regions of interest (ROIs) were defined within each subject's Talairach-space-transformed brain images using volumetric segmentation masks (Fig. 2). ROIs were chosen based on previous studies (Table 1). Initial brain tissue volume and cerebrospinal fluid (CSF) volume during normoxia were estimated with SIENAX[52], which is part of FSL[53]. Measures were normalized for skull size.

*Sodium mapping.* Sodium MR data of the whole brain were acquired using a double-resonant ($^1$H/$^{23}$Na) quadrature birdcage coil (Rapid Biomed GmbH, Rimpar, Germany). All sodium MR sequences were based on a 3D density-adapted projection reconstruction technique[54]. These sequences measure an average tissue sodium (ATS) signal and a fluid-attenuated sodium (FAS) signal (details in Supplementary Table 6). As the FAS sequence emphasizes sodium ions with short relaxation times a weighting towards the intracellular sodium compartment is achieved[18–20], see Supplementary Table 6 for sequence details.

Sodium image reconstruction was performed offline[54], see Supplementary Table 6 for details. To reduce Gibbs ringing artifacts, a Hamming filter was applied. Participants' T1-weighted MPRAGE images were skull-stripped using the brain extraction tool (BET, part of FMRIBs Software Library FSL)[52] and served as reference. Sodium images were co-registered to this reference using an affine registration with 12 degrees of freedom as implemented in FMRIBs linear image registration tool (FLIRT, part of FSL)[55].

Tissue sodium values were normalized relative to CSF sodium signals, that is, for every participant, we divided each voxel's two sodium signal values by the corresponding mean signal of the CSF. This normalization procedure served the reduction of inter-individual variance in the sodium MR signals. Due to the normalization, sodium signal values are in arbitrary units. The normalization procedure was validated in a separate experiment that tested for influences of hypoxia on CSF sodium concentration. We analyzed the CSF sodium concentration of 13 additional, healthy individuals, who did not participate in the MR experiment[56]. There was no measurable effect of 18 h of hypoxia on CSF sodium content (normoxia: 137.87 ± 3.3 mM; hypoxia: 138.34 ± 3.2 mM) ensuring that any effects of hypoxia on the normalized sodium signals arose within the specified ROI.

*Transfer constant Ktrans as index of endothelial permeability.* DCE MRI was performed using a radiofrequency-spoiled 3D gradient-echo volumetric interpolated brain examination (VIBE) sequence to monitor the contrast media kinetics of gadobutrol in infra- and supratentorial brain tissue (details in Supplementary Table 6). Gadobutrol (dose 0.1 ml/kg bodyweight) was injected intravenously using a syringe pump during image acquisition.

DCE MR images were processed using the pharmacokinetic modeling module (PkModeling, https://www.slicer.org/wiki/Documentation/4.10/Modules/PkModeling) of 3D Slicer[57]. PkModeling is based on the Tofts model[58], which models the trans-endothelial flow of contrast agent into the extravascular space. The algorithm converts the MR signal into concentration parameters and then fits the Tofts model to the so derived curve to estimate the transfer constant $K^{trans}$. To estimate the transfer constant $K^{trans}$, we selected the two-parameter Tofts model without explicit plasma volume fraction term due to the instability in pharmacokinetic analysis and model fitting at the temporal resolution of the data (>5 s). To ensure independence of the results from the underlying model, we additionally estimated $K^{trans}$ based on Patlak's model[59] using IB DCE (Imaging Biometrics, Elm Grove, WI, USA). All $K^{trans}$ values are in min$^{-1}$ units.

$K^{trans}$ was determined for every voxel and individual $K^{trans}$ maps were referenced to the corresponding T1-weighted MPRAGE images using the same affine registration as applied to sodium images.

*Diffusion imaging.* Diffusion imaging of infra- and supratentorial brain areas was performed using an echo-planar imaging sequence (details in Supplementary Table 6). T2 effects were mathematically removed from the diffusion-weighted images resulting in parametric ADC maps.

*Perfusion imaging.* Brain perfusion, i.e., the steady-state delivery of blood to the brain, was measured in the supratentorial parenchyma by pulsed arterial spin labeling (PASL). We applied pulsed quantitative imaging of perfusion with a single subtraction with thin section TI1 periodic saturation (Q2TIPS) and a proximal inversion with a control for off-resonance effects (PICORE) technique[60] (details in Supplementary Table 6). Relative tissue perfusion maps were generated using oxford_asl (part of FSL)[61,62]. To derive absolute cerebral perfusion maps (in ml g$^{-1}$ min$^{-1}$), we used the $M_0$ calibration image and the CSF signal as a reference to calculate the equilibrium magnetization of blood. Perfusion maps were projected into the individual T1-weighted MPRAGE standard space using the same affine registration routine as before.

*Oxygen content and cerebral delivery. Arterial oxygen content.* The total content of arterial oxygen $CaO_2$ (ml/dl) is determined by the amount of $O_2$ bound to hemoglobin and $PaO_2$. $CaO_2$ was approximated per Eq. (1), with Hb indicating the arterial hemoglobin concentration, 1.36 being the constant capturing the affinity of $O_2$ to hemoglobin, $SaO_2$ indicating the oxygen saturation in arterial blood, 0.003 being the constant describing the solubility of $O_2$ in blood, and $PaO_2$ indicating the arterial $O_2$ tension.

$$CaO_2 = Hb \times 1.36 \times \frac{SaO_2}{100} + 0.003 PaO_2 \, ml\, O_2 \, dl^{-1} \qquad (1)$$

*Cerebral oxygen delivery.* Cerebral oxygen delivery ($CDO_2$) was calculated based upon perfusion measurements and $CaO_2$ values per Eq. (2).

$$CDO_2 = CBF \times \frac{CaO_2}{100} \, ml\, O_2 \, min^{-1} \qquad (2)$$

**Statistics.** We quantified the effects of hypoxic exposure as the difference between pre- and post-exposure measurements normalized by pre-exposure values. Sodium signals as well as $K^{trans}$, perfusion, ADC, and $CDO_2$ correspond to within-ROI averages. The statistical relevance of all hypoxia-induced effects was assessed using $t$-tests against zero. Only values within ±3 standard deviations of the group mean were included, to avoid false positives due to outliers. We assessed the relations between the different measures acquired for each ROI, as well as similarities in each measure between pairs of ROIs using Pearson correlation analyses. Only values within ±2 standard deviations of the group mean were included into the correlational analysis, to avoid leverage effects. Note that a stricter outlier identification rule was applied for correlational than for comparative analyses as the former are more sensitive to extreme values than the latter.

We assessed potential medication-related confounds by comparing the effects of hypoxic exposure between medicated and non-medicated participants using unpaired $t$-tests. The sparsity of the data did not allow for an assessment of specific influences related to the dose or the time point of medication.

All $p$-values are two-sided and were corrected according to Benjamini and Hochberg's procedure[63] to compensate for alpha inflation due to the high number of performed statistical tests. All statistical analysis were conducted using R (version 4.0.3).

**Reporting summary.** Further information on research design is available in the Nature Research Reporting Summary linked to this article.

## Data availability

The data that have been generated in this study are available in the open science framework with the identifier https://doi.org/10.17605/OSF.IO/V9H87 [https://doi.org/10.17605/OSF.IO/V9H87]. Figures 3 and 4 have associated raw data, provided as source data file. A reporting summary for this article is available as a Supplementary Information file.

## Code availability

Experimental and analysis code are available online: https://doi.org/10.17605/OSF.IO/V9H87 [https://doi.org/10.17605/OSF.IO/V9H87].

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

## Author contributions

Conceptualization: A.B., P.B., K.S.; Data curation: A.B., S.B.; Formal analysis: A.B., S.B., A.H.; Funding acquisition: P.B.; Investigation: P.G., K.S.; Methodology: A.B., P.B., H.M.; Project administration: A.B., P.B., K.S.; Resources: A.M.N., M.B., H.M.; Software: A.B., S.B., A.M.N., S.H. Supervision: M.B., P.B., K.S.; Visualization: A.B., S.B.; Writing—original draft: A.B., S.B.; Writing—review & editing: A.B., S.B., P.B.

## Funding

## Competing interests

The authors declare no competing interests.
