## [Peer Review File · Nature Communications]

Exposure to 16 Hours of normobaric hypoxia induces ionic edema in the healthy brainREVIEWER COMMENTS

Reviewer #1 (Remarks to the Author):

The authors have used a novel method of measuring intracellular and extracellular sodium with MRI to address the question of the presence of edema after exposure to hypoxia induced by simulation of 4500m for 16 hours. Subjects were placed in a room with reduced oxygen and studied in a MRI under the low oxygen level experienced in the chamber. They used a method of measuring sodium that has been used in a number of the neurological diseases. They combined the brain sodium measures with a measure of blood-brain barrier permeability. The BBB method was an older method developed by Tofts et. al. which has been mainly replaced by the dynamic contrast-enhance MRI method using Patlak plots. They found an increase in brain volume without disruption of the BBB, suggesting that there was a movement of sodium between compartments. These observations are novel and will be of interest to others studying the effects of high altitude on the brain water and the BBB. It is difficult to determine how influential this report will be since only a few MRI centers are using the sodium imaging methods and they used an older method of measuring BBB permeability.

Reviewer #2 (Remarks to the Author):

lines 41-42 in Abstract

The authors write:

The exact nature and genesis of these edemas (plural) remains unresolved. But you only mention HACE?

The authors write:

development of ionic edemas during normobaric hypoxia instigated by simulated high altitude, in agreement with cascading models of edema formation.

To this reader it seems that "during normobaric hypoxia" and "instigated by simulated high altitude" are completely redundant. Please delete one of the two descriptions of the same experimental state.

Introduction

The authors are to be congratulated on an exceptionally well written introduction and description of the cascade of physiological events from cytotoxic to ionic to vasogenic edema.

Methods

line 142. That seems to be a lot of incomplete data. Please, more information. What did they not complete? Was there a possibility that these exclusions biased the interpretation of the outcomes?

lines 167. Please more information about treatment to reverse the headache. How many subjects, when, and were those person's data analyzed independently to assure that no effects of drug treatment were present? On lines 283-286 you mention there was no effect, but a little more transparency about who, what when regarding the drug treatment would be greatly appreciated. If drugs relieved the headache, but had no effect on imaging, that has implications for interpretation of the imaging findings importance to the pathophysiology.

Sorry, found the desired info starting on line 38. Please refer (see below or similar to this section in earlier sections of paper.)

You used SIENAX at baseline which is reasonable. Why not use it at 16 hrs, no repeat volumetric scans. SIENA using subjects as their own controls is a powerful tool without the insensitivity of SIENAX for longitudinal comparisons.

The analyses are well described and generally very robust. No mention is made of power calculations for finding a small difference in imaging parameters indicative of ionic edema. Especially since there was no correlation with AMS of any of the measurements, perhaps the study was underpowered to detect a difference.

Many now believe the data supporting a difference between normobaric and hypobaric hypoxia for many physiological responses. Could that be a factor? Perhaps it is worth a mention?

Discussion

The Discussion is very well written and carefully reasoned. This reviewer wonders if the paper would be better received if the emphasis was on hypoxia and the brain, instead of on AMS and the brain. It may seem like a small change in emphasis, but it better fits the observations. Hypoxia profoundly impacts the brain. For this reviewer the suspicion remains that we simply do not understand the link between these hypoxia-initiated events and the symptoms of AMS. But this is too small of a stuff to really get at those details. But the presentation of the hypothesis of a cascade from cytotoxic to ionic to vasogenic edema and the brilliant evidence to back it up deserves to be highlighted. Very exciting work.

Reviewer #3 (Remarks to the Author):

This study was aimed to characterize cerebral edema formation by the combined use of sodium and dynamic contrast enhanced (DCE) proton MRI in 23 healthy subjects after a 16-hr exposure to normobaric hypoxia (simulated altitude of about 4,500 m).

The authors present a well-designed study demonstrating some novel and interesting aspects. Nevertheless, there are several remarks potentially to be considered before a final recommendation can be made.

In the intro section, general aspects of progressing cerebral edema formation possibly also related to altitude/hypoxia exposure are discussed. However, no clear hypothesis concerning the present study has been stated. The authors may explain which novel findings they expected to see. As correctly reported, there are several studies exposing subjects to a similar level of hypoxia for various durations (2 – 32 hours) presenting somewhat divergent findings. Was it primarily to show that cytotoxic edema is followed by ionic edema in normobaric hypoxia? If yes, why would this new (but not really unexpected) finding be important?

Figure 1 is a core piece of the manuscript but should probably be modified as it pretends time-dependent allocation of AMS development to certain stages of cerebral edema formation. This is a by far too simple assumption and neither based on the present findings nor those derived from the cited studies. Serial MRI scans over the entire hypoxia exposure as done by Sagoo et al. would be necessary to (potentially) confirm the suggested sequence of (MRI derived) pathophysiological processes. The explanation of AMS development and cerebral edema formation (based on the cited studies) remains speculative because it is merely based on correlation analyses.

With regard to the studies included in figure 1, it also would make sense to differentiate between effects of normobaric and hypobaric hypoxia because some differences may exist, e.g. indicated by differences in AMS development (Roach et al., 1996; DiPasquale et al., 2016). Importantly, McGuire et al. (2014) showed that even the exposure to only nonhypoxic hypobaria was associated with subcortical white matter hyperintensities.

Methods applied seem to be sound and the findings are nicely presented. However, it should be emphasized that the findings only present a snapshot taken after 16 hours in normobaric hypoxia (primarily in young men) and provide only little information on the cascade of pathophysiological processes and edema formation during (and after) the exposure to hypoxia. In the light of the still uncertain knowledge on HACE pathophysiology it would be especially desirable to better understand progressing edema formation from cytotoxic and ionic to vasogenic edema and finally to hemorrhagic conversion.

No correlation was found between AMS severity and edema formation. Some subjects have been

treated with NSAIDs which did not affect cerebral edema formation but may have modified AMS severity. This might represent a potential explanation why AMS severity did not correlate with edema formation?!

Finally, the authors may elaborate a bit on the scientific and clinical relevance of their new findings, in particular with regard to normobaric hypoxia.

We would like to thank all reviewers for their thoughtful evaluation of the manuscript and helpful comments. Implementing the reviewers' suggestions has considerably strengthened the manuscript, we truly appreciate the time and effort the reviewers invested into this work.

Reviewer 1:

The authors have used a novel method of measuring intracellular and extracellular sodium with MRI to address the question of the presence of edema after exposure to hypoxia induced by simulation of 4500m for 16 hours. Subjects were placed in a room with reduced oxygen and studied in a MRI under the low oxygen level experienced in the chamber. They used a method of measuring sodium that has been used in a number of the neurological diseases. They combined the brain sodium measures with a measure of blood-brain barrier permeability. The BBB method was an older method developed by Tofts et. al. which has been mainly replaced by the dynamic contrast-enhance MRI method using Patlak plots.

They found an increase in brain volume without disruption of the BBB, suggesting that there was a movement of sodium between compartments. These observations are novel and will be of interest to others studying the effects of high altitude on the brain water and the BBB.

It is difficult to determine how influential this report will be since only a few MRI centers are using the sodium imaging methods and they used an older method of measuring BBB permeability.

Authors' response: *The study was designed to leverage our expertise in sodium MRI for the field's understanding of the effects of normobaric hypoxia on the healthy brain. We combined novel with traditional, widely accessible imaging techniques in our study and in our model-based predictions for the development of hypoxia-induced edema as indicated by MRI (Figure 1). As a consequence, future studies can build on our work without having access to sodium imaging themselves.*

We applied up-to-date dynamic contrast media-enhanced (DCE) MRI to characterize blood-brain-barrier function (I. 574-578; I. 216-223). To fit the so acquired data, we used the well-established Tofts model as it yields robust results over the complete range of permeability changes. Patlak's model is especially suitable for estimating small changes in blood-brain-barrier permeability from DCE MRI measurements. Patlak's model fits are best when DCE data is acquired with long scan times (10-30 minutes), a modest temporal resolution (< 60 seconds per image) and long baseline scans (1-4 minutes) according Barnes and colleges¹. However, our study was multiparametric, and, as a consequence, DCE sequence parameters differed from those for which Patlak's model is preferable (8 minutes scan time, temporal resolution of < 12 seconds per image, baseline scan of 23,6 seconds). We agree that Patlak's model underlies important research in recent years. Nevertheless, Tofts' model remains valid and still dominates the literature (ca. 1000 vs. 3000 publications since 2017 as indicated by google scholar, search terms: Tofts/Patlak blood brain barrier). Currently, Tofts' but not Patlak's model is accessible to the majority of researchers and practitioners as Tofts' model is implemented in multiple open-source toolboxes. Thus, by using Tofts' model we warrant comparability to the majority of studies.

To erase any doubts and establish comparability with all studies, we repeated our analysis based on Patlak's model (I. 587-589, Tab. S3). As with Tofts' model, no evidence emerged for a disruption of the blood-brain-barrier after 16 hours of exposure to normobaric hypoxia (I. 218, Tab. S3). Thus, independent of the pharmacokinetic model applied to estimate brain-barrier-permeability, our study reveals the presence of ionic edema after prolonged normobaric hypoxia.

Reviewer 2:

lines 41-42 in Abstract

The authors write:

The exact nature and genesis of these edemas (plural) remains unresolved. But you only mention HACE?

Authors' response: *We clarified in the revised manuscript that high altitude is one cause for hypoxia-induced edema in healthy individuals and listed for example stroke as another possible cause (l. 42-43).*

The authors write:

development of ionic edemas during normobaric hypoxia instigated by simulated high altitude, in agreement with cascading models of edema formation.

To this reader it seems that "during normobaric hypoxia" and "instigated by simulated high altitude" are completely redundant. Please delete one of the two descriptions of the same experimental state.

Authors' response: *We modified the sentence accordingly (l. 54-56).*

Introduction

The authors are to be congratulated on an exceptionally well written introduction and description of the cascade of physiological events from cytotoxic to ionic to vasogenic edema.

Authors' response: *Thank you!*

Methods

line 142. That seems to be a lot of incomplete data. Please, more information. What did they not complete?

Authors' response: *Four participants did not agree to complete the second MRI scan, two participants did not even complete the first MRI scan. Participants referred to nausea, claustrophobia, insomnia, or unforeseeable scheduling problems as reason. These drop-out rates might seem high, but one has to keep in mind that participants knew they were to be positioned deep inside the bore while wearing a neoprene mask that provided only oxygen-reduced air and all of this after 16 hours in a strenuous experiment. Data of three participants was lost due to off-resonant sodium data (FAS), which is unfortunately typical for sodium imaging. We added this information to the manuscript (l. 488-497).*

Was there a possibility that these exclusions biased the interpretation of the outcomes?

Authors' response: *Based on the clinical assessments, the dropout did not skew the sample, the LL-scores of participants who dropped out were comparable to those of participants who completed all measurements (LL-scores of the final sample: mean 6.21, range 1-9; LL-scores of the drop-out group: mean 5, range 4-7). We added this information to the manuscript (l. 493-494).*

lines 167. Please more information about treatment to reverse the headache. How many subjects, when, and were those person's data analyzed independently to assure that no effects of drug treatment were present?

On lines 283-286 you mention there was no effect, but a little more transparency about who, what when regarding the drug treatment would be greatly appreciated. If drugs relieved the headache, but had no effect on imaging, that has implications for interpretation of the imaging findings importance to the pathophysiology.

Sorry, found the desired info starting on line 38. Please refer (see below or similar to this section in earlier sections of paper.)

Authors' response: *To comply with nature policies the methods were moved to the end of the manuscript, thus, readers will now encounter the information earlier (l. 416-421 and l. 523).*

Most participants received analgetic treatment in the first hours of the study, only a few participants received Ibuprofen later on (0, 2, 6, and 7 hours before the second scan, see new supplementary Tab. S1). Given that the analgetic effect of Ibuprofen 400 lasts about 6-8 hours, at the time of the second scan the analgetic treatment should have had hardly any effect on headache in all but two participants. Consistently, at the time of the MRI scan there was no significant difference in AMS symptoms between participants who received treatment and those who did not on the group level (mean Lake Louise scores of 6.58 with analgetic treatment and 5.64 without treatment; l. 180-183; l. 416-426; Tab. S2).

You used SIENAX at baseline which is reasonable. Why not use it at 16 hrs, no repeat volumetric scans. SIENA using subjects as their own controls is a powerful tool without the insensitivity of SIENAX for longitudinal comparisons.

Authors' response: *Yes, longitudinal analyses in SIENA yield excellent results with respect to global brain changes. However, for our analysis we needed brain tissue parcellations not provided by SIENA.*

The analyses are well described and generally very robust. No mention is made of power calculations for finding a small difference in imaging parameters indicative of ionic edema. Especially since there was no correlation with AMS of any of the measurements, perhaps the study was underpowered to detect a difference.

Authors' response: *We have added information about the power analysis to the methods section (l496-497). As outlined in the Discussion (l. 429-440), the correlational analysis might be underpowered. However, the normoxia vs. hypoxia comparisons are sufficiently powered. Thus, we remain cautious in our interpretation of the non-significant correlation with AMS and base our conclusions on the contrast between normoxia and hypoxia.*

Many now believe the data supporting a difference between normobaric and hypobaric hypoxia for many physiological responses. Could that be a factor? Perhaps it is worth a mention?

Authors' response: *With respect to AMS empirical findings differ regarding the role of ambient pressure. More severe AMS was reported for hypobaric hypoxia in small studies lasting 8-9 hours^{2,3} but not in larger studies with 16 hours of exposure⁴. We address the role of ambient pressure in the revised manuscript (l. 397-413).*

Discussion

The Discussion is very well written and carefully reasoned. This reviewer wonders if the paper would be better received if the emphasis was on hypoxia and the brain, instead of on AMS and the brain. It may seem like a small change in emphasis, but it better fits the observations.

Authors' response: *We followed the reviewer's suggestion and shifted the emphasis towards the effects of normobaric hypoxia on the brain throughout the whole manuscript.*

Hypoxia profoundly impacts the brain. For this reviewer the suspicion remains that we simply do not understand the link between these hypoxia-initiated events and the symptoms of AMS. But this is too small of a stuff to really get at those details.

Authors' response: *We stress in the revised discussion that the link between hypoxia and AMS might be more complex than often assumed (l. 381-395) and focus on the impact of normobaric hypoxia on the brain throughout the manuscript.*

But the presentation of the hypothesis of a cascade from cytotoxic to ionic to vasogenic edema and the brilliant evidence to back it up deserves to be highlighted. Very exciting work.

Authors' response: *We thank the reviewer for the positive evaluation of our work.*

Reviewer 3:

This study was aimed to characterize cerebral edema formation by the combined use of sodium and dynamic contrast enhanced (DCE) proton MRI in 23 healthy subjects after a 16-hr exposure to normobaric hypoxia (simulated altitude of about 4,500 m).

The authors present a well-designed study demonstrating some novel and interesting aspects. Nevertheless, there are several remarks potentially to be considered before a final recommendation can be made.

In the intro section, general aspects of progressing cerebral edema formation possibly also related to altitude/hypoxia exposure are discussed. However, no clear hypothesis concerning the present study has been stated. The authors may explain which novel findings they expected to see. As correctly reported, there are several studies exposing subjects to a similar level of hypoxia for various durations (2 – 32 hours) presenting somewhat divergent findings. Was it primarily to show that cytotoxic edema is followed by ionic edema in normobaric hypoxia? If yes, why would this new (but not really unexpected) finding be important?

Authors' response:

Our goal was to measure for the first time electrolyte and water shifts as well as blood-brain-barrier integrity after prolonged normobaric hypoxia and by doing so to precisely characterize edema under these conditions. We hypothesized the presence of ionic edema, as this intermittent stage of edema formation could explain divergent findings in previous studies. Ionic edema has never been investigated or even discussed in this context before, probably because ionic edema can not be identified based on proton MRI measurements used in previous studies. We added this hypothesis to the text (l. 78-84).

The finding that cytotoxic edemas (which had been reported for shorter durations of exposure) are followed by ionic edemas is novel and important. To the reviewer, it might not seem surprising given the stringent logic and predictions of cascading models of edema formation. Yet, ionic edemas are still relatively unknown and were never even mentioned in previous studies on the effects of normobaric hypoxia on the brain (reviewed in table 1). Rather, previous studies interpreted proton MRI measurements that were in agreement with both ionic and vasogenic edema (Figure 1) in favor of the latter. Our extended analysis suggests otherwise and thus proposes an important alternation to the previously assumed time course of the development of hypoxia-induced cerebral edema.

The importance of our findings goes beyond the relation between high altitude and edema. The presence of ionic edema has never been demonstrated in healthy humans. Thus, our results close the gap between observation of cytotoxic and vasogenic edema following normobaric hypoxia and provide the first in vivo evidence for a cascading evolution of edema formation. This is an important milestone for theories of edema formation based on Starling's equation. At the same time, this study shows that the experimental protocols developed in high-altitude research in combination with novel imaging techniques provide an excellent tool to study edema formation more generally, for example, following conditions such as stroke or traumatic brain injury.

We added these points to the revised manuscript (l. 462-477).

Figure 1 is a core piece of the manuscript but should probably be modified as it pretends time-dependent allocation of AMS development to certain stages of cerebral edema formation. This is a by far too simple assumption and neither based on the present findings nor those derived from the cited studies. Serial MRI scans over the entire hypoxia exposure as done by Sagoo et al. would be necessary to (potentially) confirm the suggested sequence of (MRI derived) pathophysiological processes. The explanation of AMS development and cerebral edema formation (based on the cited studies) remains speculative because it is merely based on correlation analyses.

Authors' response: *We have removed the reference to AMS from Figure 1 and stress in the revised discussion that the link between AMS and hypoxia-induced cerebral edema appears to be more complex than is sometimes assumed (l. 381-395).*

With regard to the studies included in figure 1, it also would make sense to differentiate between effects of normobaric and hypobaric hypoxia because some differences may exist, e.g. indicated by differences in AMS development (Roach et al., 1996; DiPasquale et al., 2016). Importantly, McGuire et al. (2014) showed that even the exposure to only nonhypoxic hypobaria was associated with subcortical white matter hyperintensities.

Authors' response: *We now indicate in Figure 1 and Table 1 whether the referenced study investigated the effects of normobaric or of hypobaric hypoxia. Additionally, we stress throughout that our study investigated the effects of normobaric hypoxia on the brain and discuss that additional mechanisms, for example, those described by McGuire and colleagues (2014), might be involved in the development of AMS under hypobaria (l. 397-413).*

Methods applied seem to be sound and the findings are nicely presented. However, it should be emphasized that the findings only present a snapshot taken after 16 hours in normobaric hypoxia (primarily in young men) and provide only little information on the cascade of pathophysiological processes and edema formation during (and after) the exposure to hypoxia. In the light of the still uncertain knowledge on HACE pathophysiology it would be especially desirable to better understand progressing edema formation from cytotoxic and ionic to vasogenic edema and finally to hemorrhagic conversion.

Authors' response: *The goal of Figure 1 is to visualize that our data presents a snapshot and should be interpreted in unison with previously reported measurements following prolonged exposure to hypoxia. We agree that the pathophysiology of HACE requires further investigation, also in light of the potential role of hypobaria. However, this is not trivial, longer hypoxic exposure durations and repeated MRI scans increase the risks for participants and pose additional methodological challenges regarding DCE MRI. Our study provides a crucial step on that way in that it provides in-vivo evidence for ionic edema, suggests a revision of the time course based on previous proton MRI findings, and establishes a protocol for the investigation of hypoxia-induced cerebral edema beyond the ionic stage. We added these considerations to the discussion (l. 441-453).*

No correlation was found between AMS severity and edema formation. Some subjects have been treated with NSAIDs which did not affect cerebral edema formation but may have modified AMS severity. This might represent a potential explanation why AMS severity did not correlate with edema formation?!

Authors' response: *Usually, participants received NSAIDs in the first hours of the study. Only a few participants received them later on (0, 2, 6, and 7 hours before the second scan, see new supplementary table S1). Given that the analgetic effect lasts about 6-8 hours, there should be no influence of the NSAIDs on AMS severity for all but two participants at the time of the second scan and the final AMS symptom check. Consistently, there was no significant difference in AMS symptoms between participants who received treatment and those who did not on the group level (mean Lake Louise scores of 6.58 with analgetic treatment and 5.64 without treatment; **Tab. S2**). We added this information to the revised manuscript (l. 176-185; l. 415-426).*

Finally, the authors may elaborate a bit on the scientific and clinical relevance of their new findings, in particular with regard to normobaric hypoxia.

Authors' response: *In the revised manuscript, we explicitly stress both the scientific and clinical relevance of our findings in line with the previous comments (l. 456-477).*

Normobaric hypoxia provides a unique model to study the pathophysiological processes of edema formation in an otherwise healthy population. Our results provide in vivo evidence for a cascading evolution of ionic cerebral edema characterized by Starling's equation, a theory which until now has been developed on theoretical grounds and in animal models^{5,6}. In turn, a theoretical understanding of the mechanisms of edema formation provides an excellent base for the development of treatments for the effects of hypoxia, for example, induced by stroke or traumatic brain injury, on the brain^{6,7}.

Beyond this important theoretical contribution, our study demonstrates practically how modern imaging techniques can be used to precisely characterize the current stage of edema formation in theoretical and clinical research.

Moreover, our study ties into the literature on the development of cerebral edema under conditions of simulated high altitude, i.e., given prolonged normobaric hypoxia. Our finding of ionic edemas, an interim stage between cytotoxic and vasogenic edemas, after prolonged hypoxia bridges the gap between reports of cytotoxic and vasogenic edema in previous studies, corrects the assumed time course of edema formation in healthy humans exposed to normobaric hypoxia while revealing the need for a precise characterization of edema type through an assessment of electrolyte distributions and blood-brain-barrier permeability in future studies. At the same time, our results highlight the complexity of the relation between AMS and hypoxia-induced cerebral edema and the need to differentiate between both in research and practice.

Literature:

1. Barnes SR, Ng TSC, Montagne A, Law M, Zlokovic BV, Jacobs RE. Optimal acquisition and modeling parameters for accurate assessment of low K_{trans} blood-brain barrier permeability using dynamic contrast-enhanced MRI. *Magnetic resonance in medicine* **75**, 1967--1977 (2016).
2. Roach RC, Loeppky JA, Icenogle MV. Acute mountain sickness: increased severity during simulated altitude compared with normobaric hypoxia. *Journal of Applied Physiology* **81**, 1908-1910 (1996).
3. DiPasquale DM, Strangman GE, Harris NS, Muza SR. Hypoxia, Hypobaric, and Exercise Duration Affect Acute Mountain Sickness. *Aerospace Medicine and Human Performance* **86**, 614-619 (2015).

4. Schommer K, Menold E, Subudhi AW, Bärtzsch P. Health risk for athletes at moderate altitude and normobaric hypoxia. *British Journal of Sports Medicine* **46**, 828 (2012).
5. Simard JM, Kent TA, Chen M, Tarasov KV, Gerzanich V. Brain oedema in focal ischaemia: molecular pathophysiology and theoretical implications. *The Lancet Neurology* **6**, 258--268 (2007).
6. Stokum JA, Gerzanich V, Simard JM. Molecular pathophysiology of cerebral edema. *Journal of Cerebral Blood Flow & Metabolism* **36**, 513--538 (2016).
7. Walcott BP, Kahle KT, Simard JM. Novel Treatment Targets for Cerebral Edema. *Neurotherapeutics* **9**, 65-72 (2012).

REVIEWER COMMENTS

Reviewer #1 (Remarks to the Author):

The authors have addressed all of my concerns, and the manuscript is improved. I have no further concerns.

Reviewer #2 (Remarks to the Author):

Thanks to the authors for an excellent revision of this most interesting paper. I think this will make a major contribution to this field.

Reviewer #3 (Remarks to the Author):

The authors responded adequately to most of my comments. There is one remaining aspect needing further consideration:

The summary of previous studies on normobaric hypoxia and cerebral edema provided in table 1 is not complete. For instance, the normobaric hypoxia study by Mairer and colleagues (PMID: 23226263) is lacking. This study did not find any MRI-evidence for (cytotoxic) cerebral edema formation after an 8-hour exposure to normobaric hypoxia, which may refute a bit the "perfect" course of edema formation demonstrated in figure 1.

Again, we would like to thank all reviewers for their thoughtful evaluation of the manuscript and helpful comments - we truly appreciate the time and effort the reviewers invested into this work.

Reviewer 1:

The authors have addressed all of my concerns, and the manuscript is improved. I have no further concerns.

***Authors' response:** We very much appreciate your efforts and time to strengthen the paper.*

Reviewer 2:

Thanks to the authors for an excellent revision of this most interesting paper. I think this will make a major contribution to this field.

***Authors' response:** We thank the reviewer for the very positive evaluation of our work.*

Reviewer 3:

The authors responded adequately to most of my comments. There is one remaining aspect needing further consideration:

The summary of previous studies on normobaric hypoxia and cerebral edema provided in table 1 is not complete. For instance, the normobaric hypoxia study by Mairer and colleagues (PMID: 23226263) is lacking. This study did not find any MRI-evidence for (cytotoxic) cerebral edema formation after an 8-hour exposure to normobaric hypoxia, which may refute a bit the "perfect" course of edema formation demonstrated in figure 1.

***Authors' response:** We included the study by Mairer and colleagues in Table 1, Figure 1 and the revised manuscript (l.103-107, l.317). Given that the study simulated higher altitudes than other studies with comparable exposure durations, it fits well with our framework that the transformation from cytotoxic to ionic edema.*

REVIEWERS' COMMENTS

Reviewer #3 (Remarks to the Author):

Agreed. I do not have further comments.